# Qualitative study of in-kind incentives to improve healthcare quality in Belize: Is quality work better than wealth?

**Diego Rios-Zertuche** [1]*, **Angel Eugenio Benitez Collante** [2], **Ana Mylena Aguilar Rivera** [3], **Armelle Gillett** [4], **Natalia Largaespada Beer** [4], **Julio Sabido** [4], **Karla Schwarzbauer** [5]

**1** Salud Mesoamerica Initiative, Inter-American Development Bank, N.W., Washington, DC, United States of America, **2** Cornell University, Ithaca, New York, United States of America, **3** Inter-American Development Bank, Mexico City, Mexico, **4** Ministry of Health and Wellness, Belmopan, Cayo, Belize, **5** Salud Mesoamerica Initiative, Tegucigalpa, Honduras

* diegori@iadb.org

**Data Availability Statement:** All relevant data are within the manuscript and its Supporting information files.

## Abstract

### Background

There is a sparsity of knowledge of the specific mechanisms through which financial and non-financial incentives impact the performance of health teams. This study aims to address this knowledge gap by examining an in-kind incentives program for healthcare teams implemented in three districts in Belize (2012–2022) as part of the Salud Mesoamerica Initiative, which aimed to improve healthcare quality.

### Methods

We performed a qualitative study to understand the mechanisms through which the in-kind incentive program supported quality improvement in Belize. We conducted key informant interviews (April—June 2021) remotely on a sample of former and current healthcare workers from Belize's Ministry of Health and Wellness familiar with the program. We analyzed responses using qualitative content analysis. We used open coding to identify patterns and themes.

### Results

We conducted eight key informant interviews from a pool of thirty potential informants. Our analysis of the interviews yielded a total of 11 major themes with 27 subthemes. Most informants reported that in-kind incentives were not the primary motivation for improving their performance, though they did acknowledge that incentives had increased their attention on the quality of care provided. Conversely, we found that quarterly measurements and supportive supervision by national level authorities offered an external validation mechanism and instilled frontline staff with a sense of shared responsibility towards improving their performance. The majority of informants conveyed positive opinions about the in-kind incentives program.

**Funding:** This work was supported by the Bill & Melinda Gates Foundation [Grant Number OPPGH5328], the Carlos Slim Foundation, and the Government of Canada through the Salud Mesoamerica Initiative. Funders had no role in the design, data collection and analysis, decision to publish, or preparation of the manuscript.

**Competing interests:** The authors have declared that no competing interests exist.

## Conclusions

Our study contributes to the understanding of how in-kind incentives can enhance performance. We found that in-kind incentives created extrinsic motivation, leading to an increased focus on quality. Standardized measurements and supportive supervision improved intrinsic motivation and formed a stronger commitment to quality of care. Rather than focusing on tangible incentives, explicitly incorporating standardized measurements and supportive supervision in the routine work of the Ministry of Health could have longer lasting effects on quality improvement.

## Introduction

Almost one third of all excess deaths in low- and middle-income countries can be attributed to poor quality of care [1]. Improving quality of care is a growing priority to improve health outcomes [2]. Incentives have shown to have positive effects on team performance in healthcare settings, ultimately leading to an improvement in quality [3]. Nonetheless, the mechanisms that motivate health teams to improve quality are not fully understood [3, 4]. Furthermore, there is also debate about the effectiveness of incentives in motivating performance and determining who should be the recipient of such incentives.

Financial incentives based on productivity have been found to have positive short-term effects, particularly by enhancing extrinsic motivation [3, 5]. Extrinsic motivation, which stems from external factors such as financial rewards and verbal recognition, can be enhanced by these incentives [5]. However, research suggest that such incentives may lead to unintended long-term consequences [5]. They can create a transactional approach to performance and undermine intrinsic motivation [6, 7], which is characterized by the drive to engage in a behavior for its own sake and because it is meaningful, such as having empathy for patients [5]. In contrast, financial incentives based on capitation, which reward healthcare providers for meeting the healthcare needs of a specific population, have been found to mitigate some of these unintended effects and improve the quality outcomes [7, 8].

Non-financial incentives, such as peer-pressure and monitoring, improving teamwork, leadership and training, may enhance intrinsic motivation and lead to long-term behavior change [5, 7]. Specifically, supportive supervision has been found to improve intrinsic motivation and quality of care [9, 10]. The efficacy of pay-for-performance interventions on quality of care is strongly influenced by their design, with quality adjustment metrics and a combination of output and target payments having the greatest effect [11].

Since 2011, the Salud Mesoamerica Initiative (SMI) supported the Ministry of Health and Wellness (MOHW) of Belize, as well as other Ministries of Health in the Mesoamerican region, with the goal of improving maternal and child health for the poorest populations. SMI followed a results-based financing model, whereby an incentive award was provided to the country at the conclusion of each 2-year phase, provided that the country achieved previously agreed targets for coverage and quality of care indicators. Targets were externally verified by an independent evaluator.

In this context, the MOHW implemented the Quality Innovation Fund (QIF), which aimed to incentivize health teams to improve quality of care across health facilities [12]. The QIF offered in-kind incentives to health teams in the form of goods or services, such as equipment, training, or social activities. Incentives were complemented by technical assistance for collaborative learning, supply chain management and renewal of the community health platforms.

At the start of each year, health teams wrote a proposal with the desired goods or services up to a preestablished amount based on the facility-type and population served—ranging from $350 for health centers serving under 2,500 people to $2900 for regional hospitals. A set of five to six quality indicators and one supply indicator and targets were established for health teams in hospitals and rural health centers. Specific indicators were determined for ambulatory facilities and hospitals. Every three months, a MOHW central level representative verified availability of supplies and conducted standardized medical record reviews [13] to assess progress towards targets. Measurement criteria was explicit and known by facility staff. Quality improvement teams in hospitals, urban clinics, and larger ambulatory facilities monitored their progress monthly using the same indicators and criteria. The results for the QIF were computed for all facilities and discussed in a quarterly meeting. If the minimum score for all indicators and targets was achieved, the goods requested on the proposal were procured and delivered to the facility.

The QIF was implemented in tandem with SMI in Belize. The QIF aimed to complement the national-level incentive by focusing on the same indicators and targets. SMI covered three districts in Belize, namely Corozal, Orange Walk, and Cayo, and its two initial phases were implemented between 2013 and 2017. External measurements conducted within this timeframe have shown substantial improvements in various health indicators, such as contraception post-delivery which increased from 4.8% to 90.3%, postpartum care for women within 7 days of birth, which increased from 41.7 to 75.4%, and management of neonatal complications, which increased from 23.7% to 53.5%. Additionally, SMI improved oxytocin prophylaxis after birth from 60.0% to 98.7%, routine newborn care with quality from 30.2% to 88.7%, enrollment of newborns to child health services within 7 days from 25.3% to 66.5% and diarrhea treatment with oral rehydration salts and zinc from 20.0% to 95.3% [14].

The final phase of the SMI began in 2018 and ended in 2022, after a two-year extension due to the COVID-19 pandemic. During the final phase, incentives under the QIF were introduced in late 2021, although measurements and technical assistance continued through this period. External measurement results of this phase are still pending. While an impact evaluation is necessary to directly attribute the effects of the QIF on healthcare quality, a qualitative study can provide valuable insights into the program's functioning and inform the future design and implementation of incentive programs for healthcare quality improvement.

In this study, we seek to investigate the underlying mechanisms by which the QIF has contributed to improving healthcare quality in Belize. We conducted key informant interviews to gain a deeper understanding of what worked and what did not work, and what could be improved on future incentive programs.

## Materials and methods

### Sampling

The study followed a qualitative study design. We obtained a list of potential key informants from the MOHW including current or recent staff who had worked in SMI target areas and had knowledge and experience with the QIF. We selected a purposeful sample representing the national and regional level and health providers. We asked informants to provide the name of an additional potential informant to use for snowballing. Our target sample size was 10 with up to 5 additional informants if saturation was not achieved with the initial sample. Our criterion for saturation was hearing repeated themes during interviews. AEBC contacted potential key informants by email and telephone or WhatsApp to request the interviews. The email contained a summary of the study and informed consent (see S1 Data). AEBC scheduled interviews with those who responded. For those that did not respond, we sent at least three follow-up messages.

## Data collection

AEBC (MD, male, graduate student) conducted all interviews through audio conferences using Zoom or by telephone. Between April and June 2021, he interviewed informants individually at the time and place of their preference. AEBC received a 2-hour training on interviewing techniques and conducted two mock interviews to complement training and to pilot the instruments. AEBC had no financial or professional interest in the QIF, as he was not involved in its design or implementation and had no prior experience with incentive programs. His primary motivation was to gain a deeper understanding of hospital efficiency and healthcare quality. The findings of the study and the potential continuation or discontinuation of the QIF did not result in any personal benefit or employment opportunities for any of the researchers involved in this study.

The questionnaire consisted of eight questions (see S1 Checklist): five open-ended questions to learn about the informant's experience with the QIF, their perspective on the effect of the QIF on quality of care, the incentive scheme, indicators and measurement; one question to inquire about three proposed changes to the QIF; one multiple choice question to help determine the role of the informant, and a final question to ask about additional potential informants.

Informed consent was obtained orally before each interview from all participants. Interviews were recorded, transcribed and anonymized by AEBC. AEBC had no prior knowledge of the participants prior to the study, and only interacted with them in advance for the purpose of scheduling the interviews. Each interview lasted approximately 20 minutes. Participants could stop the interview at any time and did not receive compensation. AEBC took notes during interviews, which assisted transcribing. We did not conduct repeat interviews and did not request feedback from participants. The study was reviewed and approved by the Institutional Review Boards at Cornell University (Ithaca, NY) (Protocol ID#: 2103010245) and at Belize's MOHW (Ref. GEN/147/01/21(12) Vol. V).

All the data utilized for the analyses has been included in S1 Data; however, complete interview transcripts cannot be shared to protect informant confidentiality.

## Analysis

AEBC and DRZ conducted a collaborative analysis of the data using qualitative content analysis. We systematically categorized the transcribed data to classify, summarize, and tabulate it. We employed an open coding approach, manually assigning codes to the data, enabling us to identify patterns and themes in an inductive manner, without relying on any preexisting framework. To perform the analysis, we employed a methodology akin to the one suggested in the existing literature [15], utilizing Spreadsheet Software (MS Excel 365, Seattle, Microsoft) for qualitative content analysis. We recorded quotes from interviews in the spreadsheet and assigned each to a separate row. All responses were analyzed collectively, without considering the specific questions that were answered. In a neighboring column, the researchers provided a descriptive preliminary code per quote. Through an iterative process, similar codes were merged until no further merging was possible.

Once all the data was coded, we organized it into major themes and subthemes. To organize the codes, considering the emerging themes, the authors utilized categories from the Model for Organizational Performance and Change [16], and from the QIF programmatic components to define in themes and subthemes.

AEBC and DRZ coded and analyzed the data separately and discussed differences in their interpretation. All researchers reviewed the coded data. Findings were not discussed with informants. We followed the Consolidated criteria for reporting qualitative research [17] to present our results.

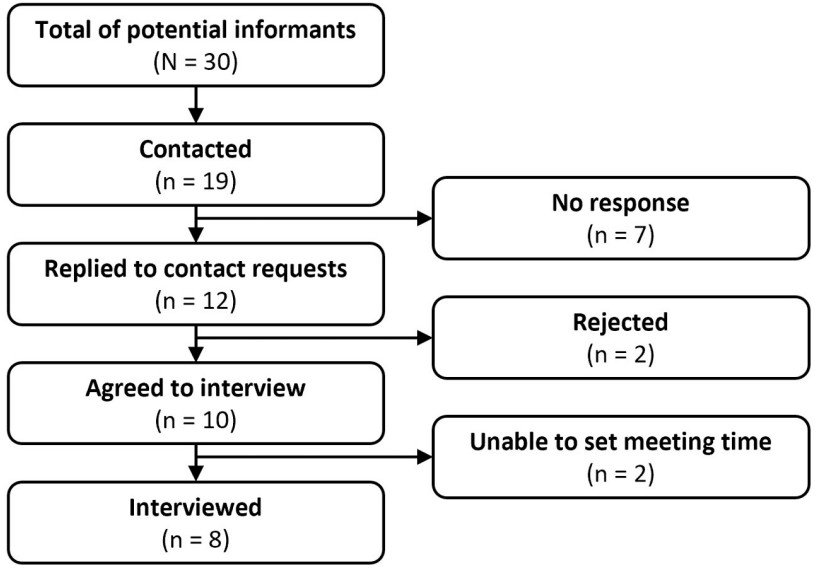

**Fig 1. Sample selection flowchart.**

## Results

We conducted a total of 8 interviews, and the final sample size is summarized in Fig 1. Two of the informants who responded the contact request declined the interview, one cited being too busy, while the other needed authorization from her supervisor. Two additional informants agreed to participate but, due to their time constraints, we could not interview them. The characteristics of informants are described in Table 1. The majority of informants were medical doctors or nurses at hospitals, with one informant at regional-level manager and another working in a rural clinic. However, two of the informants from hospitals were nurse managers of several rural clinics.

After completing coding, we organized themes and subthemes identified through open coding under three topics: collective and inter-organizational issues related to organization and performance of the MOHW; individual and Inter-personal considerations regarding individual's and work unit's motivations, values, and skills; and programmatic issues which include the QIF design and its components. We identified a total of 11 major themes and 27 sub-themes (see Fig 2).

**Table 1. Informant characteristics.**

| Role | Location | Gender |
|---|---|---|
| Manager | Regional | Female |
| Medical doctor | Hospital | Female |
| Nurse | Hospital | Female |
| Nurse | Hospital | Female |
| Nurse | Hospital | Female |
| Nurse | Rural Clinic | Female |
| Nurse Manager | Hospital | Female |
| Nurse Manager | Hospital and District | Female |

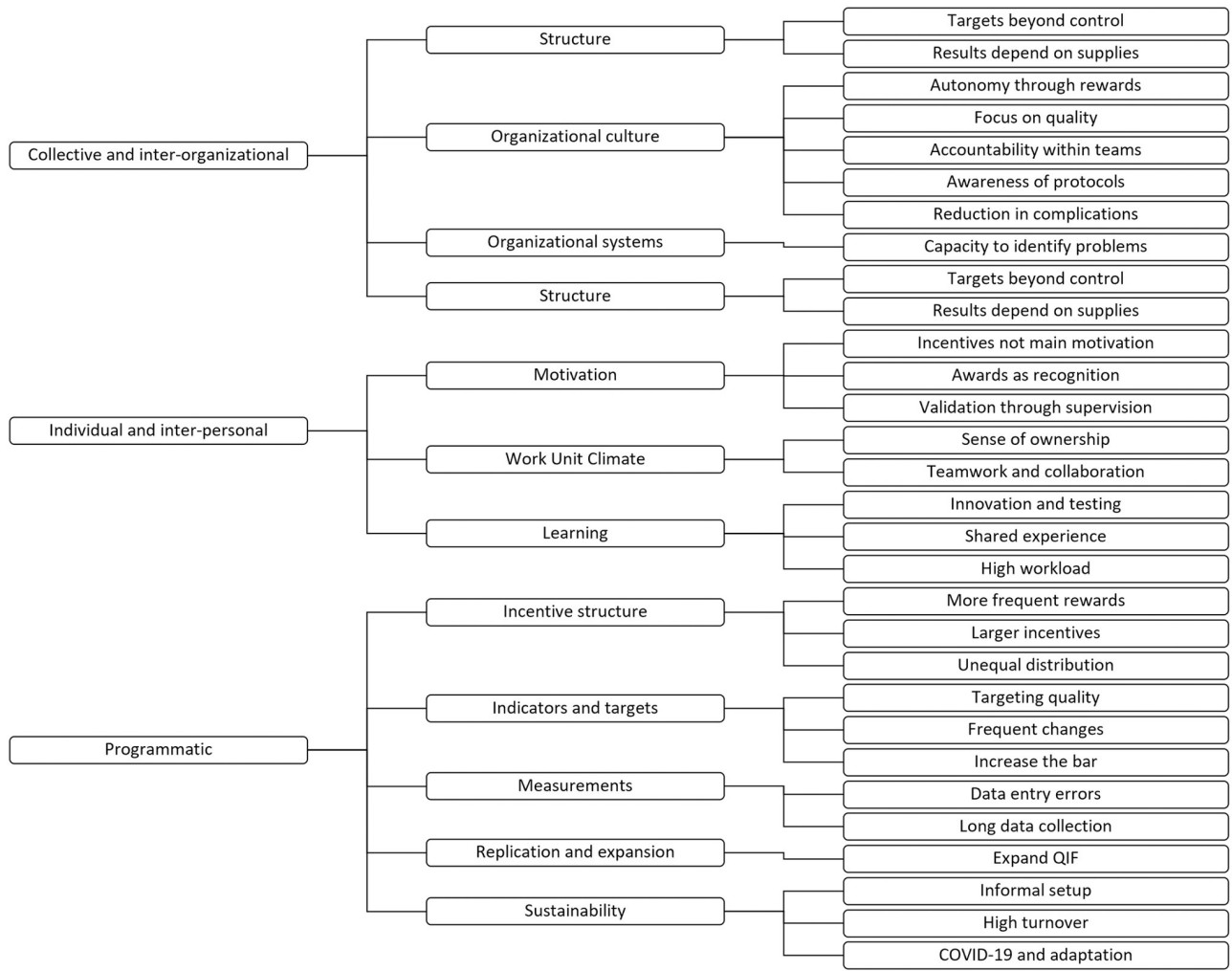

**Fig 2. Coding tree with major themes, subthemes and codes.**

## Collective and inter-organizational themes

Informants described a shift in culture in the MOHW. Most of the informants perceived that the QIF increased the focus on quality of care, not just by raising awareness of clinical practices but also by increasing accountability within teams. For example, one informant explained how nurses in their hospital worked together with doctors to ensure adherence to treatment protocols.

> *"So, whenever they would send the patient to the doctor, they will say, when you're finish, come and when they come back [. . .] [and] the treatment according to the protocol is not given, the nurse will take it upon herself and go back to the doctor and [. . .], change it so there was improvement and closer collaboration between the nurse and the doctor."*

> —*Informant 1*

Staff autonomy was facilitated by the possibility to choose the QIF award. This possibility transferred the responsibility for achieving results to the health team, thereby addressing their perceived resource limitations.

*"It was somebody asking you what do you want, what are the things that you want, what are the things that you would like to have that you don't have [. . .] Most of the time when you've been rewarded or recognized for your hard work, It's just a gift that's given to you, rather than you selecting something that would benefit you the most in the workplace [. . .]."*

—Informant 2

Informants reported that the frequent measurements helped improve organizational systems by creating the capacity to identify problems in healthcare delivery processes. The measurement results prompted discussions to analyze root causes and propose potential solutions.

*"And it comes to mind, obstetric complications, and how the monthly audits have helped us to identify problems in PPH (postpartum hemorrhage). Why the bleeding? What's the cause of the bleeding, [. . .] if different measures that we're doing were it, were done enough to ensure that the bleeding can stop, and that it will not be repeated."*

—Informant 8

Nevertheless, informants also described important structural problems that limited their capacity for improvement. Several informants described targets that were set beyond the facility's individual control. Additionally, they identified limitations in the availability of supplies and equipment as one of the main barriers to achieving results. Since frontline health service providers do not directly manage resources, their capacity for improvement is limited to what they can achieve with the available resources.

*"So, it's kind of difficult to implement any kind of intervention that requires equipment, that requires materials [. . .] we have to basically depend on the changes that [. . .] the healthcare worker can bring."*

—Informant 7

## Individual and inter-personal themes

Most informants agreed that in-kind incentives did not serve as the main motivation. Rather, many informants described genuine care for their patients, which was shared by them and their colleagues. They viewed awards were seen as a recognition for their hard work.

*"They really cared [. . .] about their patients getting the service. You know, but it wasn't really about the award itself. [. . .] Of course, they were all thankful, whenever they got anything [. . .]. But that wasn't really their motivation to work towards [achieving the] indicator itself."*

—Informant 6

The national level's close support and supportive supervision provided frontline teams external validation and motivation. The role of the national level changed, from holding health providers accountable for results to sharing responsibility with bidirectional accountability. Service providers were held accountable to the national level, and supervision visits made the national level accountable to service providers.

*"[A]nd the initiative along with the QIF give the staff an opportunity to have somebody from higher up there with them [. . .], to do the follow up, so to push, to motivate and, it felt good to have somebody from the national level come in, and that can actually sit with them and listen*

*to their concerns, and, you know, share ideas and how they can improve and how they can go doing better."*

—Informant 2

Continued measurements improved the work unit climate and increased awareness of the quality of care, prompting health providers to take ownership of their work. They recognized that it was their responsibility ensure patients received care according to standards. This commitment fostered increased teamwork and collaboration, a providers realized that providing quality care was not solely the responsibility of individuals, but rather a collective effort across departments (for example, from a patient arriving in the emergency department moving to maternity).

*"I think it's because it has given health care providers a bit of sense of ownership in terms of the quality of the care that is being offered to patients. I mean if there is a continuous checking and assurance, or constant inspection of their work. I think it is creating, it's building in them some type of ownership, making sure that they keep in track with what they're supposed to be doing what is expected of them more than, more than anything else. . ."*

—Informant 7

The QIF played an important role in creating an enabling environment for learning and innovation. Informants reported being able to continuously experiment, testing new approaches and studying the results. The shared experiences, group discussions and supervision visits facilitated dissemination of lessons and ideas from other health facilities, promoting cross-learning.

*"it created that dynamic of best practices like; okay so I tried this and, it didn't work out this way but I was able to tweak it and then I got better results when I tweaked it, or the next facility that's it okay I adapted your idea but for my community I had to change it a little bit because of maybe the size or the culture of people that are there, and it created that sharing capacity within the health facility."*

—Informant 2

However, the overall capacity of health teams and their workloads hindered innovation. An informant explained that even when they were successful in implementing new ideas, innovating was challenging when they had no spare time.

Nevertheless, the overall capacity of health teams and their workload hindered innovation. An informant explained that even when they were successful implementing new ideas, innovating was challenging when they had no spare time.

*"It didn't work completely like it wasn't a perfect method, because then you can have different services, but then again, it's one nurse providing all the services. . ."*

—Informant 6

## Programmatic themes

Informants commented on several advantages and limitations of the QIF. Concerning the incentive structure, the most significant issue was that incentives were not always available to everyone involved, particularly in hospitals. The awards primarily targeted the maternity and pediatrics departments, while maternal and child health services and support services (such as

laboratory and pharmacy) were provided across multiple hospital departments. As a result, the award created unnecessary frictions between staff in these cases.

> "So, it's a holistic effort. So, at the end of the day when you're going to give me that money, it's not for one area but it includes the entire hospital, and that is what has been creating little frictions here and there. . ."
>
> —Informant 8

Informants suggested alternative incentive structures, such as more frequent and larger awards, and focusing the awards on resources to improve the quality of care. Many informants viewed awards as a potential mechanism to address resource limitations for patient care.

Two informants commented on indicators and targets. One informant expressed frustration with the frequent changes to indicator criteria, which required staff to modify their processes accordingly. Moreover, it was not always clear how the criteria were utilized to calculate indicators. In contrast, another informant explained that it was necessary to set higher standards by introducing more challenging criteria, given that they had been using the same indicators and criteria for four years.

> [I]in terms of the criteria, I think we have some that are straight to the point, which is good. But I think at this level we have been with the same period for more than four or five years now. I think we can step it up a level, because we are getting a little bit complacent. . ."
>
> —Informant 7

Several informants reported that electronic tools had facilitated measurements. The tools guided them in reviewing medical records and collecting the specific parameters required for indicators. The data was automatically transmitted to a web server, and the results became readily available. However, the informants pointed out that they could not correct any errors once the data had been recorded, which was a significant disadvantage. Furthermore, one informant explained that the data collection for one indicator was time-consuming because too many parameters had to be collected.

> "[F]or example, and the doctor is new, [. . .] so the doctor enters incorrect something [. . .] in the tablet doesn't allow you to make any changes. . ."
>
> —Informant 5

Two informants emphasized the potential to expand the QIF to other regions of the country or to other healthcare services. They believe that several components of the QIF could increase staff motivation and improve the quality of care. Specifically, one informant mentioned that continuous measurement could serve as a mechanism for improvement.

> "Not that we're perfect, but it has helped us [. . .] identify our problems and to see exactly how good or how bad we're doing. And it would be something good to motivate the other areas of the hospital, so they can also start to realize you know what at the end of the day, at the end of two months, end of three months or on a monthly basis they will audit, what we do and to see where we're at, and how we can improve our care to the patients."
>
> —Informant 8

Many informants discussed the strengths and weaknesses of the program's sustainability. They mentioned that the program's informal onboarding and training practices, as well as its adaptability allowed for continuity. Despite changes in priorities and increased workload due to the COVID-19 pandemic, self-monitoring measurements were still being carried out. However, these mechanisms are fragile, and the high staff turnover without effective onboarding was identified as a significant threat to the program's sustainability.

*"Looking at it now and also, it's adaptive. So, they've come up with a way to still be able to auto monitor even though, all five or 10 people can't be in a [one] room due to COVID [19], but they're still evaluating themselves and everybody is still monitoring the indicators in the back of their head."*

—Informant 2

## Discussion

Our study sheds light into how incentives may affect intrinsic and extrinsic motivations of healthcare staff. We found that the QIF improved the organizational culture of the MOHW, staff motivation and the work unit climate. The informants overwhelmingly agreed that in-kind incentives were not the primary motivation. While these incentives created extrinsic motivation to obtain the reward, they were also perceived as inequitable. In contrast, quarterly measurements and supportive supervision were deemed effective in improving intrinsic motivation.

In the context of incentive programs for health, our study makes several contributions. Firstly, it elucidates the operational mechanisms through which in-kind incentives effectively enhance performance. Criticisms have been directed at incentive schemes for their excessive focus on short-term outcomes without adequate consideration for their systemic effects [18]. Therefore, it is crucial to emphasize the importance of implementing incentive schemes effectively, as the process itself plays a significant role [18]. Secondly, it highlights the key design elements of the incentive program that were important in achieving positive results. In low- and middle-income countries (LMICs), there remains a lack of understanding regarding the specific mechanisms through which incentive schemes influence health worker performance [19]. Thirdly, it introduces a new layer of discussion by exploring the potential of non-financial incentives in improving intrinsic motivation. While there is a growing body of evidence showing the positive impact of incentives on the healthcare quality [18, 20], most studies primarily compare incentive schemes with the status quo or other financing methods [18]. Few studies have focused on interventions that incorporate the main components of incentive programs, such as routine measurements, data-sharing systems, and supervision, without providing incentives.

Our findings are in-line with other studies that underline the positive effects of incentives on performance [3, 5]. Incentives were found to increase healthcare workers' attention to quality of care, which is also consistent with literature [21]. Yet, we also uncovered potential pitfalls that may lead to waning effects. Previous research has shown that perceptions of fairness in the distribution of incentives can influence their effectiveness [22]. While incentives often target physicians and other clinical staff, they may overlook other essential roles to provide quality care, such as cleaning, laboratory, and pharmacy staff. To design effective incentive schemes, it is important to consider the broader health system context [19]. Future incentive designs may need to target a wider range of people from interrelated services, particularly when aiming to improve quality of care.

Informants strongly agreed that in-kind incentives were not the main motivation. On the contrary, they highly valued quarterly measurements by the national level and self-

measurements by Quality Improvement teams. These measurements increased their intrinsic motivation by providing external validation, fostering shared responsibility, and strengthening personal competence. Previous research has found that supportive supervision can improve motivation [9]. Although supportive supervision was not an explicit feature of the QIF, most of its elements were incorporated organically [23], such as regular measurement visits, shared actionable metrics, crosspollination of lessons, and focus on quality improvement. Intrinsic motivation may have more lasting effects than extrinsic motivation [3]. In the future, the MOHW may consider an alternative QIF design without in-kind incentives, while maintaining the other prominent features and making a deliberate effort to establish supportive supervision, and to incorporate clinical staff and support staff feedback in the selection process of in-kind incentive to promote greater equity and increased participation.

Furthermore, research has shown that increasing healthcare workers' autonomy can increase intrinsic motivation [24]. The MOHW in Belize could explore ways to involve healthcare workers more in regular decision-making processes, without associating them to incentives. Different strategies have been used successfully to increase healthcare worker autonomy [25, 26] such as establishing clinical advisory committees to prequalify products during procurement processes [27], or utilizing end-users' feedback through scenario-based simulations to quality products [28]. To start, Belize could update its drug formulary with the direct participation of medical specialists from the different health regions. Nevertheless, this may be a contentious issue in many countries, such as Belize, where donors and international organizations play an important role in providing in-kind donations through their own procurement processes, which could also incorporate feedback mechanisms.

Our study had several limitations. Firstly, we were unable to fully separate the effects of SMI from the QIF. As the QIF was implemented within the framework of SMI, it might have influenced the views and perspectives of some participants. Secondly, we faced challenges in reaching our target sample size and could not reach many potential participants. While a larger sample could have provided a more comprehensive perspective, the level of agreement among respondents already highlights the most significant features of the QIF. Ultimately, we conducted interviews with over one-fourth of all potential participants. Thirdly, our study was conducted nearly four years after the last incentive was given, which may have led participants to omit important details or experiences. Nevertheless, we managed to obtain diverse and detailed accounts from all participants. Finally, it is possible that participants who agreed to participate may have had a more positive view of the QIF. However, given that the study was conducted remotely during the pandemic and noncompletion was primarily due to workload, we do not consider this to be a significant concern. It is worth noting that our research team consisted of individuals from diverse backgrounds, providing a range of views that enhanced our analysis and offered a broader perspective.

## Conclusions

Our study contributes to the understanding of how in-kind incentives can enhance performance. Our findings show that the introduction of in-kind incentives fosters extrinsic motivation among healthcare providers, leading to an increased focus on improving the quality of care. Moreover, we identified key design elements of the incentive program that effectively support performance improvement. In order to promote equity and foster greater participation, the MOHW could explore alternative designs of the QIF by incorporating these elements without the in-kind incentives or by involving healthcare workers and support staff more actively in the incentive selection process. Furthermore, we observed that standardized measurements and supportive supervision not only improved intrinsic motivation but also created

a stronger commitment to delivering quality care. Therefore, explicitly integrating these activities into the routine operations of the MOHW could have a longer-lasting effect on quality improvement as opposed to incentives. Additionally, granting healthcare workers greater autonomy, independent of incentives, has the potential to enhance their performance. Ultimately, quality work may be better than wealth after all.

## Supporting information

**S1 Checklist. COREQ (COnsolidated criteria for REporting Qualitative research) checklist.**
(PDF)

**S1 Data.**
(ZIP)

## Acknowledgments

We would like to thank the key informants that contributed their insights to this study.

## Author Contributions

**Conceptualization:** Diego Rios-Zertuche, Ana Mylena Aguilar Rivera, Armelle Gillett, Natalia Largaespada Beer, Julio Sabido, Karla Schwarzbauer.

**Data curation:** Diego Rios-Zertuche, Angel Eugenio Benitez Collante.

**Formal analysis:** Diego Rios-Zertuche, Angel Eugenio Benitez Collante.

**Investigation:** Angel Eugenio Benitez Collante.

**Methodology:** Diego Rios-Zertuche, Angel Eugenio Benitez Collante.

**Project administration:** Diego Rios-Zertuche, Ana Mylena Aguilar Rivera, Armelle Gillett, Natalia Largaespada Beer, Karla Schwarzbauer.

**Resources:** Ana Mylena Aguilar Rivera, Armelle Gillett, Julio Sabido, Karla Schwarzbauer.

**Supervision:** Diego Rios-Zertuche, Natalia Largaespada Beer.

**Validation:** Karla Schwarzbauer.

**Writing – original draft:** Diego Rios-Zertuche, Angel Eugenio Benitez Collante.

**Writing – review & editing:** Ana Mylena Aguilar Rivera, Armelle Gillett, Natalia Largaespada Beer, Julio Sabido, Karla Schwarzbauer.

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
