## [Decision Letter · Decision Letter 0]

13 Mar 2023

PONE-D-22-31837Qualitative Study of In-Kind Incentives to Improve Healthcare Quality in Belize Is quality work better than wealth?PLOS ONE

Dear Dr. Diego,

Thank you for submitting your manuscript to PLOS ONE. After careful consideration, we feel that it has merit but does not fully meet PLOS ONE’s publication criteria as it currently stands. Therefore, we invite you to submit a revised version of the manuscript that addresses the points raised during the review process.

ACADEMIC EDITOR:Please include the COREQ checklist and address the queries on the methods from the reviewers. A major revision to the manuscript is required. 

We look forward to receiving your revised manuscript.

Kind regards,

Dr. Candice Maylene Chetty-Makkan, MA, PhD

Academic Editor

PLOS ONE

Journal Requirements:

2. In the ethics statement in the Methods, you have specified that verbal consent was obtained. Please provide additional details regarding how this consent was documented and witnessed, and state whether this was approved by the IRB

3. Please include your tables as part of your main manuscript and remove the individual files. Please note that supplementary tables (should remain/ be uploaded) as separate "supporting information" files

Additional Editor Comments:

Dear Diego,

The manuscript that you submitted has value. However, the reviewers have several concerns with the methodology and approach that was used. For these reasons I am supporting a major revision.

Please consider the reviewer's suggestions and re-submit.

Kind regards,

Dr. Candice Chetty-Makkan

Reviewers' comments:

Reviewer's Responses to Questions

**Comments to the Author**

1. Is the manuscript technically sound, and do the data support the conclusions?

Reviewer #1: Yes

Reviewer #2: Partly

2. Has the statistical analysis been performed appropriately and rigorously? 

Reviewer #1: N/A

Reviewer #2: No

3. Have the authors made all data underlying the findings in their manuscript fully available?

Reviewer #1: No

Reviewer #2: Yes

4. Is the manuscript presented in an intelligible fashion and written in standard English?

Reviewer #1: Yes

Reviewer #2: Yes

5. Review Comments to the Author

Reviewer #1: There are a number of items from the COREQ guidelines which have not been included in the manuscript in it's current form. These should be included in the revision.

1. On lines 21-23 of the background, the authors should not reveal the results of the study by indicating that improvements were observed but instead they should specify what the gap in the literature is or the problem statement as well as the aim of the study.

2. Line 24, 25 and 30. Please avoid using the passive voice “was performed”, “were conducted” and “were identified”. Instead use the active voice by starting the sentence with the words “we performed”, “we conducted” and “we identified”.

3. Line 37 of the conclusion. The authors need to make clear which activities they are referring to here as this paragraph is about in-kind incentives.

4. Line 48 – please correct typing error. Should read as “middle-income”.

5. Line 113 – please write all the initials of the author ABC as it is not clear if you are referring to AEBC.

6. The authors should make it clear whether or not they established a relationship with the participants prior to study commencement. They also need to specify the interviewers characteristics such as any bias, assumptions, reasons and interests in the research topic. This is one of the requirements needed to publish qualitative research according to the COREQ guidelines.

7. Line 121 – please clarify if the two mock interviews were conducted as part of the pilot testing of the questionnaire.

8. As part of the data collection section, please specify if repeat interviews were carried out and if so, how many.

9. Also in the data collection section, please include if field notes were made or not.

10. In addition, authors should include if transcripts were returned to the participants for comment or correction.

11. The authors should explain how they assessed data saturation.

12. In the analysis section, the authors should specify which software (if any) was used for the analysis. If manual coding was performed, this should be stated.

13. The authors need to state whether or not the participants provided feedback on the findings.

14. Line 144 – 145. The word “the” is missing between “of” and “informants” in the sentence that reads: “Nevertheless, two of informants from hospitals…”

15. Line 150-151. The authors state that under the three tropics, 11 major themes and 27 sub-themes were identified. More clarity should be provided as this statement seems to imply that deductive coding was used yet the authors had previously mentioned that they used open coding (line 134). The authors should make it clear whether they simply used the topic headings to classify themes found through open coding or it they used the topic headings to find themes and patterns.

16. Lines 245-246 need to be deleted as they are repeated on lines 247-248.

17. Line 246. The word “were” in the phrase “criteria were used” seems out of place and should be omitted.

Reviewer #2: The authors report to valuable work in understanding the elements which contributed to the quality improvement within the QIF supported programs. The manuscript is very well written and is is grammatical correct.

Major reasons for rejection:

The methodology followed is not to standard and this is the main reason for the rejection, which does not comply with the Qualitative study reporting guidelines required by PLoS One. In particular, there is no that authors used the Consolidated criteria for reporting qualitative research (COREQ) checklist or Standards for reporting qualitative research (SRQR) checklist. If this was used, further detail is required and deviations to this checklist noted.

Within the Data Collection section, while there is mention of the IRB approval, no indication if consent was requested of Key Informants, what was contained in the consent process and how was this conducted. Further no indication if participants were reimbursed for their time, and if not this ought to be mentioned.

Minor comments:

1. The program was conducted between 2013 and 2017, while the key informant interviews were conducted between April-June 2021. There is no indication for the reasons for the lengthy period between program ending and the data collection. This could be a limitation in recall bias and should be acknowledged in the limitations section.

2. The reason in-depth interviews with staff working within the program was not considered? I believe some of the limitations noted of the study could have been resolved by their inclusion. Further this would have prevented the mainly high level view of the incentives provided.

3. A description of the interview guide was provided. More adequate would have been to provide the actual guide in the appendices of the manuscript submission.

4. A description of the data analysis is given, however no indication if this was manually done or with the aid of software tool.

5. While Table 1 provides the themes which emerged from interviews, there is little indication of how these themes related to one another to one another and/or if these themes coincide with any particular theoretical framework

6. the definition of intrinsic and extrinsic motivation factors is poorly defined. Since this is a major distinction and has underlying differentiation to the overall conclusion in the study, a better description with examples within the program could be indicated.

7. Lines 315-319, present a limitation of the study and is related to point 2 above where the inclusion of staff may have added value to gaining the views of both staff and management teams.

8. With a bit more description in the results section, figure 1 may not be needed.

6. PLOS authors have the option to publish the peer review history of their article (what does this mean?). If published, this will include your full peer review and any attached files.

Reviewer #1: No

Reviewer #2: No

---

## [Author Response · Author response to Decision Letter 0]

28 Mar 2023

"Comment: 1. Is the manuscript technically sound, and do the data support the conclusions?

Reviewer #1: Yes

Reviewer #2: Partly"

Response: We express our gratitude to both reviewers for their constructive comments. We have taken their feedback into consideration and made appropriate revisions, with the aim of enhancing the overall quality of our manuscript. 

"Comment: 2. Has the statistical analysis been performed appropriately and rigorously? 

Reviewer #1: N/A

Reviewer #2: No"

Response: Given that this was a qualitative study, we did not conduct statistical analyses. 

"Comment: 3. Have the authors made all data underlying the findings in their manuscript fully available?

Reviewer #1: No

Reviewer #2: Yes"

Response: We have now included the codified dataset as supplementary materials. However, we cannot share the deidentified transcripts due to the potential compromise of informant confidentiality.

"Comment: 4. Is the manuscript presented in an intelligible fashion and written in standard English?

Reviewer #1: Yes

Reviewer #2: Yes"

Response: Thank you, we reviewed the manuscript again to ensure proper use of the English language.

Comment: Reviewer #1: There are a number of items from the COREQ guidelines which have not been included in the manuscript in it's current form. These should be included in the revision.

Response: Thank you for taking the time to read our manuscript and provide constructive comments. We have revisited the COREQ guidelines and made an effort to provide more precise descriptions.

Comment: 1. On lines 21-23 of the background, the authors should not reveal the results of the study by indicating that improvements were observed but instead they should specify what the gap in the literature is or the problem statement as well as the aim of the study.

"Response: We modified the abstract to include the knowledge gap and aim of the study.

Background: There is a sparsity of knowledge of the specific mechanisms through which financial and non-financial incentives impact the performance of health teams. This study aims to address this knowledge gap by examining an in-kind incentives program for healthcare teams implemented in three districts in Belize (2012―2022) as part of the Salud Mesoamerica Initiative, which aimed to improve healthcare quality

Lines 20-24 (clean version)"

Comment: 2. Line 24, 25 and 30. Please avoid using the passive voice “was performed”, “were conducted” and “were identified”. Instead use the active voice by starting the sentence with the words “we performed”, “we conducted” and “we identified”.

Response: We revised the abstract and manuscript to use active voice.

Comment: 3. Line 37 of the conclusion. The authors need to make clear which activities they are referring to here as this paragraph is about in-kind incentives.

"Response: Thank you for pointing this out. We have edited the Conclusion to make it clearer:

Rather than focusing on tangible incentives, explicitly incorporating standardized measurements and supportive supervision in the routine work of the Ministry of Health and Wellness could have longer lasting effect than quality improvement.

Lines 20-24 (clean version)"

Comment: 4. Line 48 – please correct typing error. Should read as “middle-income”.

"Response: Corrected. 

Line 51 (clean version)"

Comment: 5. Line 113 – please write all the initials of the author ABC as it is not clear if you are referring to AEBC.

Response: We changed the initials to AEBC throughout the manuscript. 

Comment: 6. The authors should make it clear whether or not they established a relationship with the participants prior to study commencement. They also need to specify the interviewers characteristics such as any bias, assumptions, reasons and interests in the research topic. This is one of the requirements needed to publish qualitative research according to the COREQ guidelines.

"Response: We clarified that a relationship was not established before the study as follows:

AEBC had no prior knowledge of the participants prior to the study, and only interacted with them in advance for the purpose of scheduling the interviews.

Lines 158-159 (clean version)

We also added additional information to describe with more detail the interviewers characteristics:

AEBC (MD, male, graduate student) conducted all interviews through audio conferences using Zoom or by telephone. […] AEBC had no financial or professional interest in the QIF, as he was not involved in its design or implementation and had no prior experience with incentive programs. His primary motivation was to gain a deeper understanding of hospital efficiency and healthcare quality. The findings of the study and the potential continuation or discontinuation of the QIF did not result in any personal benefit or employment opportunities for any of the researchers involved in this study.

Lines 141-149 (clean version)"

Comment: 7. Line 121 – please clarify if the two mock interviews were conducted as part of the pilot testing of the questionnaire.

"Response: We added the following clarification in the text:

AEBC received a 2-hour training on interviewing techniques and conducted two mock interviews complement training and pilot the instruments. 

Lines 143-144 (clean version)"

Comment: 8. As part of the data collection section, please specify if repeat interviews were carried out and if so, how many.

"Response: No repeat interviews were carried out. We added the following clarification in the text: 

We did not conduct repeat interviews and did not request feedback from participants.

Lines 161-162 (clean version)"

Comment: 9. Also in the data collection section, please include if field notes were made or not.

"Response: No repeat interviews were carried out. We added the following clarification in the text: 

AEBC took notes during interviews, which assisted transcribing.

Lines 161 (version with changes)"

Comment: 10. In addition, authors should include if transcripts were returned to the participants for comment or correction.

"Response: We added the following clarification in the text: 

We did not conduct repeat interviews and did not request feedback from participants.

Lines 161-162 (clean version)"

Comment: 11. The authors should explain how they assessed data saturation.

"Response: We included the following text:

Our criterion for saturation was hearing repeated themes during interviews.

Lines 134-135 (clean version)"

Comment: 12. In the analysis section, the authors should specify which software (if any) was used for the analysis. If manual coding was performed, this should be stated.

"Response: We included the following clarification:

The researchers manually coded using open coding to identify patterns and themes based on the data.

Lines 167-168 (clean version)"

Comment: 13. The authors need to state whether or not the participants provided feedback on the findings.

"Response: We added the following clarification in the text: 

We did not conduct repeat interviews and did not request feedback from participants"

Comment: 14. Line 144 – 145. The word “the” is missing between “of” and “informants” in the sentence that reads: “Nevertheless, two of informants from hospitals…”

Response: Corrected. 

Comment: 15. Line 150-151. The authors state that under the three tropics, 11 major themes and 27 sub-themes were identified. More clarity should be provided as this statement seems to imply that deductive coding was used yet the authors had previously mentioned that they used open coding (line 134). The authors should make it clear whether they simply used the topic headings to classify themes found through open coding or it they used the topic headings to find themes and patterns.

"Response: Topic headings were used to organize themes and subthemes after coding was completed. We added the following clarification:

After completing coding, we organized themes and subthemes identified through open coding under three topics Lines 181-183 (clean version)"

Comment: 16. Lines 245-246 need to be deleted as they are repeated on lines 247-248

Response: Eliminated repeated text.

Comment: 17. Line 246. The word “were” in the phrase “criteria were used” seems out of place and should be omitted.

"Response: Rephrase the sentence as follows: 

Moreover, it was not always clear how the criteria were utilized to calculate indicators.

Line 297 (clean version)"

Comment: Reviewer #2: The authors report to valuable work in understanding the elements which contributed to the quality improvement within the QIF supported programs. The manuscript is very well written and is is grammatical correct.

Response: Thank you for your feedback. We have made further improvements to our editing and have addressed the issues you raised.

Comment: Major reasons for rejection: The methodology followed is not to standard and this is the main reason for the rejection, which does not comply with the Qualitative study reporting guidelines required by PLoS One. In particular, there is no that authors used the Consolidated criteria for reporting qualitative research (COREQ) checklist or Standards for reporting qualitative research (SRQR) checklist. If this was used, further detail is required and deviations to this checklist noted.

Response: In line with your suggestion, and the comments of the other reviewer, we have added more details to comply with the COREQ and SRQR checklists. 

Comment: Within the Data Collection section, while there is mention of the IRB approval, no indication if consent was requested of Key Informants, what was contained in the consent process and how was this conducted. Further no indication if participants were reimbursed for their time, and if not this ought to be mentioned.

"Response: Thank you for pointing this out. We have added the following clarifications: 

Informed consent was obtained orally before each interview from all participants. […] Participants could stop the interview at any time and did not receive compensation. […] The study was reviewed and approved by the Institutional Review Boards at Cornell University (Ithaca, NY) (Protocol ID#: 2103010245) and at Belize’s MOHW (Ref. GEN/147/01/21(12) Vol. V).

Lines 157-164 (clean version)"

Comment: Minor comments: 1. The program was conducted between 2013 and 2017, while the key informant interviews were conducted between April-June 2021. There is no indication for the reasons for the lengthy period between program ending and the data collection. This could be a limitation in recall bias and should be acknowledged in the limitations section.

"Response: Thank you. We have clarified that some components (measurements) continued, although the latest incentive before our study was in 2017. We have included recall bias as a limitation as follows:

Third, we conducted our study almost four years after the last incentive was given, which could have created recall bias and the omission of important details.

Lines 384-385 (clean version)"

Comment: 2. The reason in-depth interviews with staff working within the program was not considered? I believe some of the limitations noted of the study could have been resolved by their inclusion. Further this would have prevented the mainly high level view of the incentives provided.

"Response: As stated in our manuscript, all informants worked within the program. The implementation of the program in Belize was carried out by making functional adjustments within the Ministry of Health and Wellness, rather than contracting separate staff. See:}

The majority of informants were medical doctors or nurses at hospitals, with one informant at regional-level manager and another working in a rural clinic. However, two of the informants from hospitals were nurse managers of several rural clinics. 

Lines 177-178 (clean version)"

Comment: 3. A description of the interview guide was provided. More adequate would have been to provide the actual guide in the appendices of the manuscript submission.

Response: Thank you for this suggestion. We have now included the interview guide in the Annex B. 

Comment: 4. A description of the data analysis is given, however no indication if this was manually done or with the aid of software tool.

"Response: We included the following clarification:

The researchers manually coded using open coding to identify patterns and themes based on the data.

Lines 167-168 (clean version)"

Comment: 5. While Table 1 provides the themes which emerged from interviews, there is little indication of how these themes related to one another to one another and/or if these themes coincide with any particular theoretical framework

"Response: We derived the themes through the open coding process and did not rely on any specific framework. To make them more accessible, we categorized them into three topics. We added the following clarification in the text:

After completing coding, we organized themes and subthemes identified through open coding under three topics Lines 181-183 (clean version)"

Comment: 6. the definition of intrinsic and extrinsic motivation factors is poorly defined. Since this is a major distinction and has underlying differentiation to the overall conclusion in the study, a better description with examples within the program could be indicated.

"Response: Thank you for your suggestion. We have expanded the description of intrinsic and extrinsic motivation in our manuscript:

Extrinsic motivation, which stems from external factors such as financial rewards and verbal recognition, can be enhanced by these incentives [5]. However, research suggest that such incentives may lead to unintended long-term consequences [5]. They can create a transactional approach to performance and undermine intrinsic motivation [6,7], which is characterized by the drive to engage in a behavior for its own sake and because it is meaningful, such as having empathy for patients [5].

Lines 60-65 (clean version)"

Comment: 7. Lines 315-319, present a limitation of the study and is related to point 2 above where the inclusion of staff may have added value to gaining the views of both staff and management teams.

"Response: We agree that a larger sample may have added a more detailed perspective. However, as we describe in our manuscript, the sample does include staff with management and health service provision perspectives. Note that in Belize, due to staffing limitations, many people take multiple roles. See:

The majority of informants were medical doctors or nurses at hospitals, with one informant at regional-level manager and another working in a rural clinic. However, two of the informants from hospitals were nurse managers of several rural clinics. 

Lines 181-183 (clean version)"

Comment: 8. With a bit more description in the results section, figure 1 may not be needed.

Response: Thank you for your valuable suggestion! We appreciate your input. We would like to highlight that Figure 1 is a helpful addition to the manuscript as it presents a concise summary of the results, which is also recommended in the COREQ checklist. We believe that Figure 1 will be useful for readers who are looking for a quick overview of the research findings.

---

## [Decision Letter · Decision Letter 1]

11 May 2023

PONE-D-22-31837R1Qualitative Study of In-Kind Incentives to Improve Healthcare Quality in Belize: Is quality work better than wealth?PLOS ONE

Dear Dr. Rios-Zertuche,

Thank you for submitting your manuscript to PLOS ONE. After careful consideration, we feel that it has merit but does not fully meet PLOS ONE’s publication criteria as it currently stands. Therefore, we invite you to submit a revised version of the manuscript that addresses the points raised during the review process.

We look forward to receiving your revised manuscript.

Kind regards,

Dorothy Lall

Academic Editor

PLOS ONE

Additional Editor Comments:

Thank you for your revisions in response to the comments. Please respond carefully to the reviewers comments that request you to report in accordance with the COREQ guidelines. Perhaps a checklist can be appended in response to the comment. I would also recommend having a qualitative method expert advise you on the analysis that is presented. The coding process needs to be described in more detail and the use of a framework would make the analytic process more robust. Recall bias and risk of bias are not meaningful constructs in the qualitative approach. In fact in keeping with the approach please comment on data saturation, reflexivity and truth value of the research. Table 1 is also not in keeping with the qualitative approach and is not required instead a description of the participants is needed.

The manuscript would benefit from English language editing as pointed out by reviewer 1.

Please also reference in the discussion the debate regarding pay for performance ..the jury is still hung and the evidence of benefit is not convincing.

We look forward to a revision that considers these issues highlighted.

Reviewers' comments:

Reviewer's Responses to Questions

**Comments to the Author**

1. If the authors have adequately addressed your comments raised in a previous round of review and you feel that this manuscript is now acceptable for publication, you may indicate that here to bypass the “Comments to the Author” section, enter your conflict of interest statement in the “Confidential to Editor” section, and submit your "Accept" recommendation.

Reviewer #1: (No Response)

Reviewer #2: (No Response)

2. Is the manuscript technically sound, and do the data support the conclusions?

Reviewer #1: Yes

Reviewer #2: Partly

3. Has the statistical analysis been performed appropriately and rigorously? 

Reviewer #1: N/A

Reviewer #2: N/A

4. Have the authors made all data underlying the findings in their manuscript fully available?

Reviewer #1: Yes

Reviewer #2: Yes

5. Is the manuscript presented in an intelligible fashion and written in standard English?

Reviewer #1: Yes

Reviewer #2: Yes

6. Review Comments to the Author

Reviewer #1: Minor comments

1. Line 41 (clean version): Please confirm if the word “than” is a typo in the statement “…explicitly incorporating standardized measurements and supportive supervision in the routine work of the Ministry of Health and Wellness could have longer lasting effect than quality improvement.” I’m not clear if the authors meant to say “on”.

2. Please check grammar on line 144 (clean version) that reads “…conducted two mock interviews complement training and pilot the instruments.” I’m not sure if the authors meant to say “…two mock interviews to complement training…”?

Reviewer #2: The authors have taken the time to respond to each of the queries raised and have paid attention to some of the fundamental concerns raised in the previous review. Thank you. I am though at this stage not able to fully confirm that the COREQ guidelines have been followed to the 32 point scale. I would ask that the COREQ 32 point checklist is included in a revised submission, allowing to compare the text included to the checklist, and assessing if analysis was conducted to standard. Further, I am assuming the COREQ standard is applied as this was not indicated in methods along with the appropriate reference. If the Standards for reporting qualitative research (SRQR) checklist was used, this should likewise be included as an appendix and noted in text to indicate the standard applied.

7. PLOS authors have the option to publish the peer review history of their article (what does this mean?). If published, this will include your full peer review and any attached files.

Reviewer #1: No

Reviewer #2: No

---

## [Author Response · Author response to Decision Letter 1]

17 Jun 2023

Editorial Comments

Authors' Response: Thank you for your valuable suggestions. We have incorporated important revisions into our manuscript, aiming to provide a more comprehensive and robust perspective on our results.

Comment: Thank you for your revisions in response to the comments. Please respond carefully to the reviewers comments that request you to report in accordance with the COREQ guidelines. Perhaps a checklist can be appended in response to the comment.

Authors' Response: As suggested by you and one of the reviewers, we have appended the COREQ checklist to our manuscript. See supplementary materials. 

Comment: I would also recommend having a qualitative method expert advise you on the analysis that is presented. The coding process needs to be described in more detail and the use of a framework would make the analytic process more robust. 

Authors' Response: Thank you. We expanded the description of the coding process and included details of the reference framework used, as follows: 

 “AEBC and DRZ conducted a collaborative analysis of the data using qualitative content analysis. We systematically categorized the transcribed data to classify, summarize, and tabulate it. We employed an open coding approach, manually assigning codes to the data, enabling us to identify patterns and themes in an inductive manner, without relying on any preexisting framework. To perform the analysis, we employed a methodology akin to the one suggested in the existing literature [15], utilizing Spreadsheet Software (MS Excel 365, Seattle, Microsoft) for qualitative content analysis. We recorded quotes from interviews in the spreadsheet and assigned each to a separate row. All responses were analyzed collectively, without considering the specific questions that were answered. In a neighboring column, the researchers provided a descriptive preliminary code per quote. Through an iterative process, similar codes were merged until no further merging was possible.” (page 9, lines 169-178)

Comment: Recall bias and risk of bias are not meaningful constructs in the qualitative approach. In fact in keeping with the approach please comment on data saturation, reflexivity and truth value of the research. Table 1 is also not in keeping with the qualitative approach and is not required instead a description of the participants is needed.

Authors' Response: We removed recall bias, risk of bias from the manuscript, and replaced Table 1 with a description of participants. 

We added the following text in the discussion to comment on data saturation:

 “Although a larger sample could have provided us a richer perspective, the degree of agreement between respondents already reveals the most salient features of the QIF. In the end, we interviewed over one-fourth of all potential informants.” (Page 20, Lines 417-419)

To comment on reflexivity and truth value, we included the following: 

 “It is worth noting that our research team consisted of individuals from diverse backgrounds, providing a range of views that enhanced our analysis and offered a broader perspective.” (page 20, lines 425-427)

We also believe that the COREQ checklist, and data provided in the Supplementary files increase the credibility, dependability and auditability of our study. 

Comment: The manuscript would benefit from English language editing as pointed out by reviewer 1.

Authors' Response: We have closely reviewed the manuscript for English language use. 

Comment: Please also reference in the discussion the debate regarding pay for performance ..the jury is still hung and the evidence of benefit is not convincing.

We look forward to a revision that considers these issues highlighted

Authors' Response: Thank you, this is indeed an important highlight. We included the following paragraph in the discussion: 

 “In the context of health incentive programs, our study makes several important contributions. Firstly, it clarifies how in-kind incentives effectively enhance performance by revealing their operational mechanisms. Criticisms have been raised against incentive schemes for their excessive emphasis on short-term outcomes, neglecting their systemic effects [20]. Therefore, it is crucial to emphasize the effective implementation of incentive schemes, as the process itself plays a significant role [20]. Secondly, our study highlights the key design elements of the incentive program that were crucial in achieving positive results. In low- and middle-income countries (LMICs), there is still a lack of understanding regarding the specific mechanisms through which incentive schemes influence health worker performance [19]. Thirdly, our study introduces a new layer of discussion by exploring the potential of non-financial incentives in improving intrinsic motivation. Although there is a growing body of evidence demonstrating the positive impact of incentives on healthcare quality [17, 18], most studies primarily compare incentive schemes with the existing practices or other financing methods [17]. Only a few studies have focused on interventions that incorporate the main components of incentive programs, such as routine measurements, data-sharing systems, and supervision, without providing financial incentives.” (page 18, lines 365-378)

Reviewer 1

Authors' Response: Thank you for you time reviewing our manuscript. We appreciate your constructive comments. 

Comment: 1. Line 41 (clean version): Please confirm if the word “than” is a typo in the statement “…explicitly incorporating standardized measurements and supportive supervision in the routine work of the Ministry of Health and Wellness could have longer lasting effect than quality improvement.” I’m not clear if the authors meant to say “on”.

Authors' Response: Thank you for catching this typo. We have corrected the abstract and checked the entire manuscript (see page 3, line 43).

Comment: 2. Please check grammar on line 144 (clean version) that reads “…conducted two mock interviews complement training and pilot the instruments.” I’m not sure if the authors meant to say “…two mock interviews to complement training…”?

Authors' Response: We rephrase the sentence as follows: 

 “AEBC received a 2-hour training on interviewing techniques and conducted two mock interviews to complement training and to pilot the instruments.” (see page 8, lines 143-144)

Reviewer 2

Authors' Response: Thank you for your time reviewing our manuscript. Your suggestion helped us include a couple of items that we had previously missed. 

Comment: The authors have taken the time to respond to each of the queries raised and have paid attention to some of the fundamental concerns raised in the previous review. Thank you. I am though at this stage not able to fully confirm that the COREQ guidelines have been followed to the 32 point scale. I would ask that the COREQ 32 point checklist is included in a revised submission, allowing to compare the text included to the checklist, and assessing if analysis was conducted to standard. Further, I am assuming the COREQ standard is applied as this was not indicated in methods along with the appropriate reference. If the Standards for reporting qualitative research (SRQR) checklist was used, this should likewise be included as an appendix and noted in text to indicate the standard applied.

Thank you. We have now included the following text in our manuscript and attached the COREQ checklist as a supplementary file:

 “We followed the Consolidated criteria for reporting qualitative research [17] to present our results.” (page 10, line 188)

---

## [Editor Report · Decision Letter 2]

9 Aug 2023

Qualitative Study of In-Kind Incentives to Improve Healthcare Quality in Belize: Is quality work better than wealth?

PONE-D-22-31837R2

Dear Dr. Rios-Zertuche,

We’re pleased to inform you that your manuscript has been judged scientifically suitable for publication and will be formally accepted for publication once it meets all outstanding technical requirements.

Kind regards,

Dorothy Lall

Academic Editor

PLOS ONE

Additional Editor Comments (optional):

Thank you for your careful consideration of the comments.
---

## [Editor Report · Acceptance letter]

11 Aug 2023

PONE-D-22-31837R2 

Qualitative Study of In-Kind Incentives to Improve Healthcare Quality in Belize: Is quality work better than wealth? 

Dear Dr. Rios-Zertuche:

I'm pleased to inform you that your manuscript has been deemed suitable for publication in PLOS ONE. Congratulations! Your manuscript is now with our production department. 

Kind regards, 

on behalf of

Dr. Dorothy Lall 

Academic Editor

PLOS ONE